# hPG_80_ (Circulating Progastrin), a Novel Blood-Based Biomarker for Detection of Poorly Differentiated Neuroendocrine Carcinoma and Well Differentiated Neuroendocrine Tumors

**DOI:** 10.3390/cancers14040863

**Published:** 2022-02-09

**Authors:** Aman Chauhan, Alexandre Prieur, Jill Kolesar, Susanne Arnold, Léa Payen, Younes Mahi, Berengere Vire, Madison Sands, B. Mark Evers, Dominique Joubert, Lowell Anthony

**Affiliations:** 1Division of Medical Oncology, and the Markey Cancer Center, University of Kentucky, Lexington, KY 40536, USA; susanne.arnold@uky.edu (S.A.); lowell.anthony@uky.edu (L.A.); 2ECS Progastrin, 1004 Lausanne, Switzerland; a.prieur@ecs-progastrin.com (A.P.); y.mahi@ecs-progastrin.com (Y.M.); b.vire@eurobiodev.com (B.V.); d.joubert@ecs-progastrin.com (D.J.); 3College of Pharmacy, and the Markey Cancer Center, University of Kentucky, Lexington, KY 40536, USA; jill.kolesar@uky.edu; 4Lyon Sud Hospital, 69310 Pierre-Benite, France; lea.payen-gay@chu-lyon.fr; 5School of Medicine, University of Kentucky, Lexington, KY 40536, USA; madison.sands@uky.edu; 6Department of Surgery, and the Markey Cancer Center, University of Kentucky, Lexington, KY 40536, USA; mark.evers@uky.edu

**Keywords:** circulating progastrin, hPG_80_, blood-based diagnostic biomarker, neuroendocrine neoplasms, neuroendocrine tumors, neuroendocrine carcinoma, small-cell carcinoma

## Abstract

**Simple Summary:**

Current blood-based biomarkers for neuroendocrine neoplasms (NENs) lack both sensitivity and specificity. Human circulating progastrin (hPG_80_) can be easily measured in plasma by ELISA. This study is the first to examine hPG_80_ in NENs. The study demonstrated increased levels of hPG_80_ in all sub-types of NENs, with a high sensitivity and specificity demonstrated. Plasma hPG_80_ in NENs may be a diagnostic blood biomarker for both low- and high-grade NENs; further study is warranted. A prospective multi-center trial is ongoing in NET to evaluate hPG_80_ as a means of monitoring disease (NCT04750954).

**Abstract:**

Current blood-based biomarkers for neuroendocrine neoplasms (NENs) lack both sensitivity and specificity. Human circulating progastrin (hPG_80_) is a novel biomarker that can be easily measured in plasma by ELISA. This study is the first to examine hPG_80_ in NENs. Plasma hPG_80_ was quantified from 95 stage IV NEN patients, using DxPG_80_ technology (ECS Progastrin, Switzerland) and compared with hPG_80_ concentrations in two cohorts of healthy donor controls aged 50–80 (*n* = 252) and 18–25 (*n* = 137). Median hPG_80_ in NENs patients was 5.54 pM compared to 1.5 pM for the 50–80 controls and 0.29 pM the 18–25 cohort (*p* < 0.0001). Subgroup analysis revealed median hPG_80_ levels significantly higher than for either control cohort in neuroendocrine carcinoma (NEC; *n* = 25) and neuroendocrine tumors (NET; *n* = 70) including the small-cell lung cancer (SCLC) sub-cohort (*n* = 13). Diagnostic accuracy, estimated by AUCs, was high for NENs, as well as both sub-groups (NEC/NET) when compared to the younger and older control groups. Plasma hPG_80_ in NENs may be a diagnostic blood biomarker for both low- and high-grade NENs; further study is warranted. A prospective multi-center trial is ongoing in NET to evaluate hPG_80_ as a means of monitoring disease (NCT04750954).

## 1. Introduction

Neuroendocrine neoplasms (NENs) are heterogeneous tumors that originate from various organs and are of variable aggressiveness based on grade and morphology. The incidence of neuroendocrine tumors (NETs) has increased 6.4-fold, making NETs the second most prevalent gastrointestinal malignancies after colorectal cancer [1,2]. Currently, no reliable blood-based biomarkers have been identified for NENs. Several biomarkers have been proposed for well-differentiated NETs (e.g., chromogranin A and pancreastatin); however, to date, all of the proposed biomarkers are not particularly sensitive and/or specific [3].

Progastrin, a precursor of gastrin is synthetized by G cells in gastric antrum and later processed into gastrin [4]. In normal physiologic state, progastrin esnt accumulate in antral G cells, as compared to G34-Gly and fully matured gastrin [5]. Hence, progastrin is not detectable in the blood in normal subjects, barring few exceptions. [6]. In contrast, as a result of GAST gene expression, high hPG_80_ levels are detected in blood of cancer patients [7,8,9]. *GAST*, a target of the ß-catenin/Tcf4 pathway, is also found to be activated in many solid tumors [10]. Role of hPG_80_ in tumorigenesis has been well documented in prior publications [9,11,12,13,14]. Therefore, we conducted a pilot study to test the presence of hPG_80_ in neuroendocrine neoplasms.

## 2. Materials and Methods

### 2.1. Patients and Control Cohorts

Institutional Review Board (IRB) approval was secured and banked plasma from patients with high-grade neuroendocrine carcinoma and well-differentiated neuroendocrine tumors was accessed from the University of Kentucky Markey Cancer Center’s biospecimen repository.

Control group comprised of plasma samples from two distinct age groups. Plasma from healthy, 137 non fasting (18–25-year-old) blood donors was obtained from the French blood agency (Etablissement Français du Sang) [15]. A second cohort consisted of 50–80-year-old; (median 55). Plasma from these 252 fasting subjects was obtained from PROCODE study (NCT03775473, https://clinicaltrials.gov/ct2/show/NCT03775473, accessed on 17 December 2021).

### 2.2. hPG_80_ Level Measurements in the Blood Samples

hPG_80_ was analyzed using ELISA DxPG_80_.lab kit from ECS-Progastrin. The analytical performances of the kit are described in Cappellini et al. [16]. Limit of Detection (LoD) and the limit of Quantitation (LoQ) is at 1 pM and 3.3 pM respectively. The inter- and intra-assay coefficients of variation (CV%) is below 10%. No cross-reactivity was detected with gastrin-17, Gastrin-Gly or CTFP (C-Terminus Flanking Peptide). No cross-reactivity was detected with other blood biomarker such as CA125, CEA or PSA. No interference was detected with chemicals such as SN-38, 5-FU or triglycerides, cholesterol, or hemoglobin.

### 2.3. Statistical Analysis

Differences in hPG_80_ levels were evaluated using Mann–Whitney *U* tests. The diagnostic discriminative accuracy of hPG_80_ levels in patients with cancer compared to healthy subjects was assessed using Receiver Operating Characteristics (ROC) curve analysis. Both control groups were used for ROC curve analysis in order to obtain a range of values regarding the diagnosis value of hPG_80_. Prism software (GraphPad Prism version 9.4 for Windows, GraphPad Software, La Jolla, CA, USA, www.graphpad.com, accessed on 17 December 2021) was used to perform all the statistical analysis and to create figures. The level of significance was set at *p* < 0.05.

## 3. Results

### 3.1. Diagnostic Performance of hPG_80_ in the Various Cohorts of Cancer Patients

The demographic characteristics of the neuroendocrine neoplasm (NEN) patients and control cohorts are shown in Table 1.

The median age of NEN patients was 61 years (37–86 y). Among the 95 NENs, 25 patients (26.3%) had high-grade neuroendocrine carcinoma (NEC) and 70 patients (73.7%) were diagnosed with well-differentiated neuroendocrine tumors (NET). The median hPG_80_ in NENs patients was 5.54 pM (IQR 2.07–17.11 pM) as compared to 1.5 pM (IQR 0.60–3.09 pM) for patients in the 50–80-year-old control group and 0.29 pM (IQR 0.00–1.27 pM) for patients in the 18–25-year-old cohort (*p* < 0.0001, two-tailed Mann–Whitney *U*-test). A subgroup analysis of NENs revealed a median hPG_80_ of 3.54 pM (IQR 2.02–19.91 pM) in neuroendocrine carcinoma (NEC *n* = 25) and 5.8 pM (IQR 1.91–16.74 pM) in neuroendocrine tumor (NET *n* = 70). Interestingly, the small-cell lung cancer sub-cohort (*n* = 13) also showed significant elevation of hPG_80_ with a median at 9.09 pM (IQR 2.66–25.33 pM). All the above-mentioned differences were statistically significant as compared to healthy controls (Figure 1). Table 2 shows the levels of plasma hPG_80_ in the various study subgroups.

### 3.2. Diagnostic Performance of hPG_80_ in Each Sub-Cohort of Cancer Type

As shown in Figure 2, diagnostic accuracy, estimated by the ROC AUCs, is 0.89 for all NENs, 0.87 for NETs, and 0.92 for NECs when compared to the young 18–25 y control group; for the older 50–80 y cohort, the values were 0.75 for all NENs, 0.74 for NETs, and 0.75 for NECs. 

As shown in Figure 3, diagnostic performances of hPG_80_ were tested using 90% specificity (CI: 89.8% to 93.4% for the 18–25-year-old and 84.4% to 93.3% for the 50–80-year-old control groups) for all sub-cohorts. The sensitivity ranged from 58.67% (CI: 40.74% to 74.49%) for lung other than SCLC to 69.23% (CI: 42.37% to 87.32%) for SCLC, when compared to the 18–25 y control group. The sensitivity ranged from 37.50% (CI: 21.16% to 57.29%) for NEC to 53.85% (CI: 29.14% to 76.79%) for SCLC, when compared to the 50–80 y control group.

## 4. Discussion

Neuroendocrine neoplasms represent a group of diseases with a common neuroendocrine lineage however each subgroup maintains its unique morphology, molecular biology, and phenotype. WHO now clearly distinguishes poorly differentiated neuroendocrine carcinoma aka NEC from well differentiated neuroendocrine tumor, also known as NET. NEC is characterized by an aggressive clinical course, is distinctively poorly differentiated, displays large cell or small cell morphology, and has a molecular profile indicative of mutation in TP53 and loss of RB1. In contrast, the well differentiated NETs are relatively indolent and can be differentiated into grade 1, 2, and 3 based on Ki 67 index. It is critical to distinguish between NET and NEC as both can have a variable clinical course and management paradigms [17]. However, a common unifying factor for both NET and NEC is lack of reliable blood-based diagnostic biomarker. hPG_80_ (human circulating progastrin) is a pan tumor biomarker that has been found to be over-expressed in multiple malignancies. hPG_80_ synthesis is product of overexpression of *GAST* gene in cancer cells and human circulating progastrin can now be measured accurately with help of DxPG_80_ Enzyme-Linked Immunosorbent Assay (ELISA) based test [8,16].

In this study, we investigated the diagnostic value of plasma hPG_80_ in patients with both low- and high-grade NENs. Our data revealed that plasma hPG_80_ levels in 95 patients with NENs was significantly higher compared to healthy blood donors. NENs are a heterogenous group of diseases that can arise from any location in the body [18]. Because of its broad expression in all types of NENs, plasma hPG_80_ could represent a novel biomarker to diagnose NENs. In fact, except chromogranin A (CgA), which is secreted by most NENs, currently-used biomarkers (neuron-specific enolase, pancreatic peptide and 5-hydroxyindoleacetic acid) have restricted expression, thus limiting their diagnostic value [19]. We showed that hPG_80_ diagnostic sensitivity ranged from 58.67% to 69.23% with a 90% specificity in NENs. Only few biomarkers including CgA are currently used in the diagnosis of patients with NENs [20]. Although these biomarkers are considered useful to aid diagnosis, they display limited sensitivity and specificity. Indeed, the sensitivity and specificity of CgA ranges between 43 and 100%, and below 50%, respectively [3]. In addition, elevated CgA levels are also found in many other conditions such as renal and hepatic failure, inflammatory diseases, or following the use of proton-pump inhibitors [21]. Another limitation to the use of CgA for NENs diagnosis comes from the absence of direct involvement in the mechanisms that underlie tumorigenic process including cell proliferation, migration, invasion, and metastasis [22]. By contrast, numerous studies have demonstrated that hPG_80_ contribute to tumorigenesis and tumor development [9,11,12,13]. Plasma hPG_80_ levels correlate with tumor burden but also with tumor activity. Indeed, hPG_80_ level significantly decreases upon surgery in peritoneal carinomatosis and upon remission in hepatocellular cancer [8]. In addition, in hepatocellular cancer, the hPG_80_ level correlates with response to treatment and disease progression [8]. Last but not least, we have recently shown that hPG_80_ could be used for prognosticating survival in metastatic renal cell carcinoma [7]. In line with these data and to gain further insight into its potential utility as a biomarker in NENs, the role of hPG_80_ in disease monitoring is being studied in an NCI (National Cancer Institute) sponsored multi center NET clinical trial (ETCTN 10450). Plasma hPG_80_ will be serially collected at baseline, at each radiographic (CT/MRI) assessment, and at the time of progression.

Recently, the NETest, a blood-derived multianalyte test that measures the expression of 51 circulating mRNA in blood, has been shown to outperform CgA in the diagnosis of NENs [23]. NETest diagnostic accuracy, sensitivity, and specificity, were 97, 99, and 95%, respectively [23]. However widespread use of this biomarker may be limited [24].

Due to the rarity of NECs, most of the molecular studies have focused on NETs. They demonstrated that signaling pathways such as PI3K-Akt, Ras/Raf/MEK/ERK, Notch, and Wnt/β-catenin signaling contribute to NET pathogenesis [10]. The Wnt-signaling pathway is essential to regulate cell proliferation, migration during embryogenesis, and tumorigenesis in various cancers [25]. Gene mutations in well-known components of the Wnt-signaling pathways, such as *β-catenin* and *APC*, and repression of Wnt inhibitor genes by DNA methylation or histone modification of their promoters (*SFRP-1*, *Axin-2*, *DKK-1*, *DKK-3,* and *WIF-1)* are frequently observed in NET cell lines and human NET tumor samples [10]. In addition, analysis of somatic mutations across 21 types of NETs revealed that the gene *MEN1*, encoding a negative regulator of β-catenin, was the most commonly mutated gene in NETs (8 out of 21 NET types) [26,27]. Remarkably the gene *GAST*, encoding progastrin, is a direct target of the Wnt oncogenic pathway [28]. Therefore, we can hypothesize that abnormal expression of the Wnt signaling pathway in NENs could be linked to the overexpression of hPG_80_ in these tumors.

Our study presents some limitations starting with the retrospective nature of the study and the low patient number, especially in the different subgroups analyzed. Second, most of the patients (84%) have advanced disease (stage IV), not allowing for evaluation of the diagnostic value of hPG_80_ in early-stage patients. Lastly, the current study does not account for potential confounders like use of proton-pump inhibitors, or fasting vs. post-prandial state. We acknowledge that PPI usage could induce gastrin hyper secretion (hypergastrinemia); however, to our knowledge, there is no publication showing an increase in hPG_80_ during PPI usage. Furthermore, as described in Cappellini [16], we would like to mention that the kit (DxPG_80_) that was used to measure hPG_80_ (human circulating progastrin) does not recognize any other forms of gastrin peptides, avoiding any risk of false positivity due to PPI usage and hypergastrinemia. Nonetheless, we are currently setting up a clinical study to answer this very question. Finally, a large prospective study, including more cases with early-stage disease, should strengthens the significance of using hPG_80_ as a new diagnostic biomarker in NENs.

## 5. Conclusions

This first-ever study of plasma hPG_80_ in NENs confirms that hPG_80_ is elevated in both low- and high-grade NENs, and suggests that hPG_80_ could be considered as a potential diagnostic blood-based biomarker for NENs diagnosis.

## Figures and Tables

**Figure 1 cancers-14-00863-f001:**
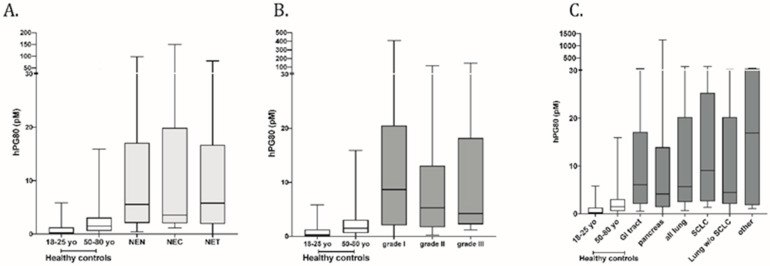
Diagnostic performance of hPG_80_ in (**A**) NENs, NEC and NET patient cohorts, (**B**) by tumor grade, (**C**) by tumor site as compared to the 18–25-year-old and 50–80-year-old.

**Figure 2 cancers-14-00863-f002:**
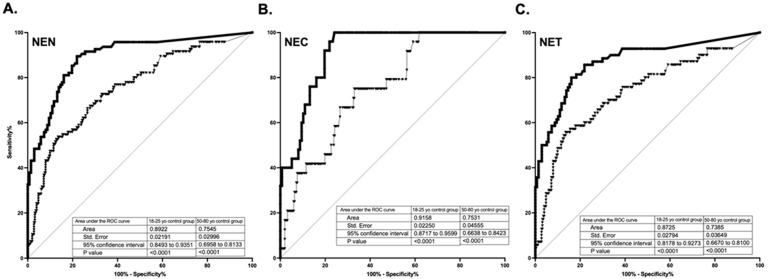
Diagnostic accuracy, estimated by the Receiver Operator Characteristic (ROC) Area Under the Curve (AUC)s, is 0.89 for all NENs (**A**), 0.92 for NECs (**B**), and 0.87 for NETs (**C**) when compared to the young 18–25 y control group (square); for the older 50–80 y cohort, the values were 0.75 for all NENs (**A**), 0.75 for NECs (**B**), and 0.74 for NETs (**C**) (triangles).

**Figure 3 cancers-14-00863-f003:**
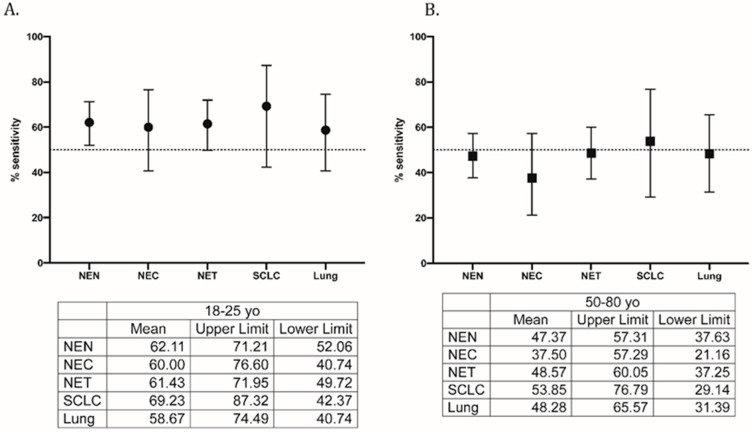
Sensitivity of hPG_80_ in the all-patient (neuroendocrine neoplasm) cohorts with a specificity set at 90% as compared to (**A**) 18–25 y control group and (**B**) 50–80 y control group.

**Table 1 cancers-14-00863-t001:** Clinical and pathological characteristics for NEN, NEC, NET patients, and control cohorts.

		NEN	NEC	NET	Control Cohorts	
					18–25 Years Old	50–80 Years Old
		N (%)	N (%)	N (%)	N (%)	N (%)
		*n* = 95	*n* = 25	*n* = 70	*n* = 137	*n* = 252
Age (years)	Median (range)	61 (37–86)	61 (37–78)	62 (37–86)	21 (18–25)	55 (50–80)
Gender	Male	38 (40%)	10 (40%)	28 (40%)	79 (57.7%)	99 (39.3%)
	Female	57 (60%)	15 (60%)	42 (60%)	58 (42.3%)	153 (60.7%)
hPG_80_	Median (IQR), pM	5.54 (0.00–1241)	3.54 (1.13–154.1)	5.8 (0.00–1241)	<LoD	<LoQ
	Mean (SD) pM	28.24 (128.8)	20.7 (39.96)	30.55 (148.5)	<LoD	3.82 (0.55)
Stage	I to III	14 (14.8%)	2 (8%)	12 (17.2%)	NA	
	IV	80 (84.2%)	23 (92%)	57 (81.4%)		
	Unknown	1 (1.0%)	0 (0%)	1 (1.4%)		
Grade	1	33	0	33		
		(34.7%)	(0%)	(47.2%)		
	2	28	0	28		
		(29.5%)	(0%)	(40%)		
	3	30	25	5		
		(31.6%)	(100%)	(7.1%)		
	Unknown	4	0	4		
		(4.2%)	(0%)	(5.7%)		
Primary Site	GI tract	46	4	42		
		(48.4%)	(16%)	(60%)		
	Pancreas	15	2	13		
		(15.8%)	(8%)	(18.6%)		
	SCLC	13	13	0		
		(13.7%)	(52%)	(0%)		
	Lung w/o SCLC	17	4	13		
		(17.9%)	(16%)	(18.6%)		
	Other	4	2	2		
		(4.2%)	(8%)	(2.8%)		

NA: not applicable; LoD: Limit of Detection; LoQ: Limit of Quantification.

**Table 2 cancers-14-00863-t002:** hPG_80_ (pM) in plasma from neuroendocrine neoplasm patients depending on the grade and the primary site. Groups are not significantly different from each other.

		GRADE						
		Grade I		Grade II			Grade III	
hPG_80_	Median (range), pM	8.66 (0.00–1241)		5.3 (0.00–142.1)			4.22 (1.13–154.1)	
	Mean (SD), pM	48.71 (214.4)		16.37 (32.1)			19.18 (36.77)	
		PRIMARY SITE						
		GI Tract	Pancreas		SCLC	Lung not SCLC		Other
hPG_80_	Median (range), pM	6.06 (0.00–142.1)	4.14 (0.00–1241)		9.09 (1.35–124.1)	4.51 (0.00–34.85)		16.89 (1.13–86.87)
	Mean (SD), pM	13.88 (24.02)	88.9 (318.9)		30.36 (52.78)	11.37 (11.35)		30.45 (39.78)

hPG_80_: circulating progastrin; IQR: interquartile range; pM: picomolar; SD: standard deviation; GI: gastrointestinal; SCLC: small-cell lung cancer.

## Data Availability

All data generated or analyzed during this study are included in this published article. Individual deidentified patient data can be shared upon request and IRB approval.

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
