# Peer review of "hPG80 (Circulating Progastrin), a Novel Blood-Based Biomarker for Detection of Poorly Differentiated Neuroendocrine Carcinoma and Well Differentiated Neuroendocrine Tumors"

_cancers, 2022, doi:10.3390/cancers14040863_

Round 1
Reviewer 1 Report
The Authors have satisfied all the requests.
Reviewer 2 Report
Dear authors,
Thank you ver much for the revised manuscript.
I am pleased and have no further objections to publication.
Reviewer 3 Report
The authors have responded to all the comments
This manuscript is a resubmission of an earlier submission. The following is a list of the peer review reports and author responses from that submission.
Round 1
Reviewer 1 Report
In this manuscript titled “hPG80 (circulating progastrin), a novel blood-based biomarker for detection of poorly differentiated neuroendocrine carcinoma (NEC) and well differentiated neuroendocrine tumors (NET)” the authors Aman Chauhan et al. evaluated the diagnostic value of Human circulating progastrin (hPG80), a novel biomarker, easily measured in plasma by ELISA, in a cohort of 95 patients with NENs subdivided into NEC (n=25) NET (n=70), compared with 2 cohorts of 389 controls subdivided into 2 subgroup according to the age. (I suggested to write this in the summary). This study is the first to examine hPG80 in NENs.
I was very happy to review such excellent paper. The results are convincing of the interest of the blood evaluation of hpG80 in grade IV NEN. However, I suggest to add some information
for the readers which are not familiar with the NEN. Would you give the main differences between NET and NEC and mention the reference of the last review of the experts of WHO classification: Rindi G et al Modern pathol 2018. It is also important to stress on the clinical interest to differentiate NET and the different stages and grades in the blood for the follow-up of the patients
about hPG80: When it has been described and who are the first to describe this marker as a cancer marker. What type of marker is it?
For increasing the proofs of the interest of this new marker:
In the table 1, I suggest to pool the patients’ tumor stages I, II, III and compared with the patients’ values of tumors stage IV to each statistical value
The mean value of hPG80 is clearly higher in NET than in the controls (please note the p value in the table). But the low values in each group are lower than the highest limit of the control of 50-80 years. Could you precise the age of such patients and comment it in the text. What no data for patients 25-50 years of the control cohorts?
Have you data for some patients about the pre and post-surgery hPG80 values and the evolution of the value with the tumor progression?
In the figure 2, there is a mistake (twice B). I suggest to put the curves of the controls in the same graph as the corresponding tumor types.
For the small mistakes, when the manuscript is as good as this one, I ask the editor to join the corrected manuscript with my review. It is easier for the authors and it avoids a second review.
Reviewer 2 Report
The Authors assess the potential role of progastrin as an oncological marker for neuroendocrine neoplasms. Biomarkers currently used (i.e., chromogranin A) have been widely investigated in several studies, but the need for new tests is currently debated. The results of the current study might be encouraging, especially if we consider the high number of healthy controls enrolled. However, a major bias of the study is the lack of data regarding the use of proton pump inhibitors (PPI) among both NEN and healthy subjects, which might significantly affect results. The analysis cannot preclude this consideration.
In addition other major comments need to be addressed:
- Authors need to clearly define when the progastrin measure was assessed in the NEN subgroup: at diagnosis? during follow-up?
- Although the study was retrospective, was the power calculation of the study assessed, in order to evaluate the number of healthy subjects needed?
- We agree with the Authors regarding the different disease stage for NENs as a potential bias for results. A sub-analysis should be added (also reported as supplementary material) showinga comparison between localized (stage I-II) and advanced (stage III-IV) disease
- All data should be reported as median (range) instead of IQR, in order to provide a more complete overview of measures distribution.
Additional comments:
- Table 1: report p-values, at least regarding the comparison of NENs vs healthy controls
- Report p-values for Fig. 1 and Table 2
- line 61: gastrin is written twice
- line 176: mistake: AUC 0.87
Reviewer 3 Report
Dear Chauhan and colleagues,
Thank you for the manuscript. I have some comments which are listed here.
- I feel that the writing of the introduction to the manuscript was a little too hasty. The ref. from JAMA Oncology by your group indicate that the increase is 6.98 fold and not only 6-fold. Further, the paper which reports these data are your 2018 data in Oncotarget and not the editorial from 2019 in JAMA Oncology. Please change the reference and the 6-fold statement in the introduction.
- The ref. 3 chosen to describe that there are no reliable biomarkers in NET reports about factors in colorectal adenocarcinoma. Is this the correct reference?
- Why is hPG80 an interesting biomarker? Are high levels of hPG80 a poor prognostic marker, is PFS lower in these patients compared to patients with low hPG80? In other words. What is the clinical significance?
- What do we know about the biomarker, is there a cut-off level?
- In Material and Methods you start the section by "IRB". What is IRB short for?
- Table 1. 70 patients with NET are included - of which 28 are male and 41 female... This does not add-up. Please look at your data again.
Reviewer 4 Report
The authors have done a fair analysis using patient samples. The following comments need to be addressed
Minor comments
Line 61: “Gastrin” is duplicated
Line 162: The authors have mentioned that the differences are statistically significant but no stats were shown in the figure.
The text/legends in figure 2 should be made visible.
